# Selecting the Appropriate Downstaging and Bridging Therapies for Hepatocellular Carcinoma: What Is the Role of Transarterial Radioembolization? A Pooled Analysis

**DOI:** 10.3390/cancers15072122

**Published:** 2023-04-02

**Authors:** Victor Lopez-Lopez, Kohei Miura, Christoph Kuemmerli, Antonio Capel, Dilmurodjon Eshmuminov, David Ferreras, Alberto Baroja-Mazo, Pedro Cascales-Campos, María Isabel Jiménez-Mascuñán, José Antonio Pons, Maria Isabel Castellon, Francisco Sánchez-Bueno, Ricardo Robles-Campos, Pablo Ramírez

**Affiliations:** 1Department of General and Digestive Surgery, Virgen de la Arrixaca University Hospital, IMIB-Arrixaca, 30120 Murcia, Spain; 2Digestive and Endocrine Surgery and Transplantation of Abdominal Organs Research Group, Biomedical Research Institute of Murcia (IMIB), 30120 Murcia, Spain; 3Department of General, Visceral and Transplantation Surgery, University Hospital Virgen de la Arrixaca. Ctra., Madrid-Cartagena, s/n, El Palmar, 30120 Murcia, Spain; 4Division of Digestive and General Surgery, Niigata University Graduate School of Medical and Dental Sciences, Niigata 950-2181, Japan; 5Department of Surgery, Clarunis—University Center for Gastrointestinal and Liver Diseases, 4052 Basel, Switzerland; 6Department of Vascular Intervenional Radiololy, Virgen de la Arrixaca University Hospital, IMIB-Arrixaca, 30120 Murcia, Spain; 7Department of Surgery and Transplantation, Swiss Hepato-Pancreato-Biliary (HPB) Center, University Hospital Zurich, 8091 Zurich, Switzerland; 8Department of Hepatology, Virgen de la Arrixaca University Hospital, IMIB-Arrixaca, 30120 Murcia, Spain; 9Department of Nuclear Medicines, Virgen de la Arrixaca University Hospital, IMIB-Arrixaca, 30120 Murcia, Spain

**Keywords:** liver transplant, hepatocellular carcinoma, downstaging, bridging, transarterial radioembolization

## Abstract

**Simple Summary:**

The usefulness of transarterial radioembolization (TARE) as a treatment has been endorsed by different authors and recommended by a multidisciplinary working group in terms of its safety, efficacy and feasibility for the surgical resection of patients with borderline-resectable hepatocellular carcinoma (HCC). In contrast, in the field of liver transplant (LT), the role of TARE is under debate and there are still no clear and unified guidelines for its indication.

**Abstract:**

Background: Transarterial radioembolization in HCC for LT as downstaging/bridging has been increasing in recent years but some indication criteria are still unclear. Methods: We conducted a systematic literature search of primary research publications conducted in PubMed, Scopus and ScienceDirect databases until November 2022. Relevant data about patient selection, HCC features and oncological outcomes after TARE for downstaging or bridging in LT were analyzed. Results: A total of 14 studies were included (7 downstaging, 3 bridging and 4 mixed downstaging and bridging). The proportion of whole liver TARE was between 0 and 1.6%. Multiple TARE interventions were necessary for 16.7% up to 28% of the patients. A total of 55 of 204 patients across all included studies undergoing TARE for downstaging were finally transplanted. The only RCT included presents a higher tumor response with the downstaging rate for LT of TARE than TACE (9/32 vs. 4/34, respectively). Grade 3 or 4 adverse effects rate were detected between 15 and 30% of patients. Conclusions: TARE is a safe therapeutic option with potential advantages in its capacity to necrotize and reduce the size of the HCC for downstaging or bridging in LT.

## 1. Introduction

Hepatocellular carcinoma (HCC) affects approximately 500,000 patients worldwide. It is the sixth most common of all cancers and the second leading cause of cancer-related death [1,2]. HCC patients used to be widely distributed particularly in East Asia and Africa. However, the incidence and mortality have recently shown a marked increase in North America and Europe making HCC a highly relevant disease worldwide [3,4]. For many years, surgical resection was the treatment of choice, but locoregional therapies and chemotherapies have been developed for complementary use. As a consequence, the survival rates of the patients treated even without curative intent have remarkably improved in this new multimodal concept [5].

Liver transplant (LT), still the ultimate treatment for HCC, is intended for the treatment of patients with advanced HCC or impaired liver function due to causative liver disease which cannot be surgically treated. Nowadays, expanded transplant indication criteria have been reported and adopted by various countries. The results are largely comparable to those of the Milan criteria which still constitute the gold standard [6], making it clear that LT is the definitive and curative treatment for HCC [7,8,9,10].

Currently, 1100 patients with HCC undergo LT annually in Europe, and the number of new cases of HCC is reported to be as high as 65,000 [11]. The management of patients with HCC during the waiting period before LT should therefore be emphasized. Surveillance and bridging therapies to ensure patients remain within transplant criteria are based on patient background, the causative liver disease of HCC, and the patient’s condition. On the other hand, a subgroup of patients with disease beyond transplant criteria are offered the same therapy but with a downstaging intent. In many cases, however, the indication and treatment procedures are center specific based on centers’ experiences.

There is no consensus for all stages, from indication criteria to the selection of treatment procedures. The role of transarterial radioembolization (TARE) using radioactive isotope, β-ray emitting Yttrium-90 in HCC treatment is still unclear. In this review, we focus on TARE which has been increasing as a locoregional therapy for HCC in recent years, in order to examine the indication criteria for LT as downstaging/bridging treatment, and its outcomes to form the basis towards future studies and treatment recommendations.

## 2. Materials and Methods

### 2.1. Database Searching and Selection Criteria

A medical librarian developed a systematic search strategy to browse PubMed, Web of Science and ScienceDirect databases, using a combination of standardized index terms and plain language to cover the terms “HCC”, “liver transplant”, “transarterial radioembolization”, “downstaging” and “bridging” as comprehensively as possible. These keywords were defined by consensus among authors and customized for each database. Key review studies were identified, and their reference lists were examined for relevant articles. The search was completed in August 2022. Refer to Figure 1 for a detailed flow diagram according to PRISMA guidelines. The study protocol was registered in PROSPERO (registration number: CRD42023383661).

To be eligible for screening, the studies had to meet the following criteria: (a) identification of TARE as a locoregional therapy for downstaging or bridging in LT as the main objective, and (b) including humans. Studies lacking adequate information on the experimental design were not included. We excluded case reports, cohorts with fewer than 3 patients, reviews, letters, commentaries, and studies published only as abstracts. Data were independently extracted by investigators using a standardized form and disagreements were resolved by consensus. Two researchers (VL & KM) independently screened bibliographies of relevant review articles and publications in the field. The same two researchers together screened titles and abstracts from the publications. In the event of disagreement, a third reviewer (CK) was involved.

### 2.2. Quality Assessment

Data were independently extracted by all investigators using a standardized form and disagreements were resolved by consensus. Study quality assessment was based on the Cochrane risk of bias tool using the following domains: sequence generation and allocation concealment; performance and detection bias; incomplete outcome data; selective outcome reporting; and other biases [12]. Understandably, it is not feasible to conduct fully blinded studies for this research question, as both the patients and staff know the nature of the intervention. Given these difficulties, if a study did not mention any blinding of staff or patients and it was not possible to contact the authors, the study was assumed to be unblinded and therefore at high risk of performance and detection bias. It was, however, possible for detection bias to be reduced by using standardized criteria for complications and discharge, and for outcome assessors to be unaware of the patient’s allocation.

### 2.3. Data Synthesis

Data synthesis was performed using narrative methods. Analyses were descriptive and displayed in tabular or graphical formats. Frequency and percentage were reported for categorical variables and median with interquartile range (IQR) were reported for continuous variables. Frequencies and survival after LT were pooled using the meta and metasurvival packages and random-effects. Survival rates and number at risk were reconstructed from Kaplan Meier estimators using Datathief III, v. 1.7 (2006).

### 2.4. TARE Procedure

Treatment commonly reported as TARE was defined as including the following procedures. In the TARE technique, microspheres impregnated with a radioisotope yttrium-90 are percutaneously and selectively delivered through the hepatic artery to targeted tumor. Yttrium-90 is a beta-ray emitter with a short half-life and limited tissue penetration. Normally, microspheres are available in glass or resin, which differ in size, activity of individual beads, and number of microspheres injected. Tumor mapping with angiography using cone-beam CT and treatment simulation with 150 MBq technetium-99 m macro-aggregated albumin are usually performed prior to treatment. After treatment, SPECT/CT or PET/CT will also be performed to assess microsphere distribution and mean dose to the target, liver, and lungs.

TARE methods include whole liver TARE, in which microsphere is spread from the proper hepatic artery to the entire liver, bilobar TARE, in which catheters are inserted into both lobes simultaneously, and lobar or segmental and subsegmental TARE, which targets one lobe or less. In addition, a two-phase TARE is performed to promote gradual atrophy of the responsible liver lobe and enlargement of the healthy liver lobe by applying the partial atrophy of the liver caused by segmental TARE (Figure 2).

## 3. Results

A total of 14 studies were included (7 downstaging, 3 bridging and 4 mixed downstaging and bridging). Patient characteristics for each study are shown in Table 1 and Table 2 and are stratified by treatment intent. The criteria used for downstaging and/or bridging in studies are depicted in Appendix A. The patient selection was based on disease factors, patient performance status and the risk of collateral damage to other organs in most reports. The different types of TARE included are depicted in Figure 2.

### 3.1. Downstaging

The proportion of whole liver TARE was between 0 and 1.6% of the studies. Multiple TARE interventions were necessary for 16.7% up to 28% of patients. The reasons for repeated interventions were remaining viable tumor in imaging and an insufficient initial procedure, respectively, or staged procedures due to bilobar disease. Seven studies reported solely on TARE to downstage HCC [13,14,15,16,17,18,19]. A total of 55 of 204 patients across all included studies undergoing TARE for downstaging were finally transplanted.

Iñarrairaegu et al. described that two patients had LT, three patients were resected and one patient had radiofrequency ablation followed by surgical resection after TARE in a series of patients with UNOS T3 disease [13]. Mehta and Dhondt et al. reported a median as high as two interventions after initial TARE to achieve downstaging to within transplant criteria [17,19]. Pracht et al. reported a series of 18 patients with ipsilateral portal vein thrombosis [14]. Two (11%) were converted to resection (*n* = 1) and transplantation (*n* = 1) after TARE, respectively. Radiologically, 2 complete and 13 partial responses (83% overall response rate) were observed. In addition, the therapy had a considerable effect on the tumor thrombus which ranged from complete involution of the thrombus in three patients and partial patency in eight patients. Overall, 13 patients (72%) showed a response in both the tumor and the thrombus.

In a series of 349 patients from the Mount Sinai Hospital [16], UNOS T2 was the transplant criterion and 22 patients of those underwent LT while the others had resections. Pathologic and radiologic complete response rates were 34% and 56%, respectively. Two studies from the University of Bologna described their experience over 11 years. In the first report, Gramenzi et al. reported their initial experience including 63 consecutively treated patients [15]. Two patients (3.2%) were downstaged after TARE and underwent LT. The complete and partial response was achieved in 46 patients (73%). Serenari et al. included 17 patients with PVT and 5 patients (29.4%) who had LT [18]. Fourteen patients (85.6%) showed a radiologic response to TARE. Among patients who had LT (*n* = 5), two had residual disease on pathological assessment. 

In a large multicenter study, Mehta and colleagues compared TARE and TACE. In the TARE group, which comprised 62 patients, only 8 patients (12.9%) had progressive disease [17]. Fourteen patients (22.6%) underwent LT and pathology showed that four had a complete necrotic tumor (30.8%) and three (23.1%) were beyond Milan criteria. In a randomized control trial, Dhondt et al. compared TARE and TACE in nonsurgical BCLC stage A and B HCC and 9 of 33 patients with TARE were downstaged to LT [19]. Recurrence was observed in 8 out of 55 patients who had LT. Survival after transplant ranged from 70.3% to 100% after one year and 80% to 91% after three years in three studies. One study reported only a one-year outcome. Gramenzi et al. reported data from a first experience which had worse survival after one year (44.7%), three years (19.0%) and five years (9.5%).

### 3.2. Bridging

Three studies reported on the effect of bridging to transplant for HCC in 74 patients [20,21,22]. Two articles only selectively described outcomes for patients who had LT [20,21]. A third study provided details of dropouts while on the waiting list [22]. Mantry et al. initially assessed 111 patients and 6 (5.4%) underwent LT after TARE and intent to bridge [20]. After 6 months, when only 43 patients were assessed, 26 showed a complete or partial response on imaging (60.5%). In this study, 65% of patients had at least one additional local or systemic treatment. In a cohort of 40 patients, 50% had multiple TARE procedures due to bilobar disease [21]. Radiologically, no patient had disease progression and in 87.5% of specimens, there was complete or partial tumor necrosis. Zori et al. reported 28 patients who had TARE [22] and 1.46 procedures were carried out per patient. All explanted livers showed some degree of necrosis, and eight patients (28.6%) had complete necrosis. Nine patients (22.5%) developed recurrence after 13 months. Median survival in two studies for patients after LT ranged from 46 to 69 months, but, in one study, median survival was not reached, and the 3-year survival rate was 92.9%.

### 3.3. Mixed Indications

Some more institutions and collaborations reported their experience with mixed intent for TARE and 141 patients who all had LT. Tohme et al. included 20 patients who had TARE and underwent LT [23]. Two patients had two TARE procedures due to bilobar disease. Six patients were outside the Milan criteria before TARE and therefore were treated for downstaging, and two (33%) were successfully converted to being within the Milan criteria (2/6). All others were within the Milan criteria and therefore had TARE as bridging therapy. The pathologic assessment revealed that only two patients had disease extent beyond the Milan criteria and therefore 66% or four patients were pathologically downstaged. Four patients had recurrence after a median of 36.8 months and the median survival was 75.1 months. Abdelfattah and colleagues presented a case series including nine patients, three had downstaging and six had bridging to transplant [24]. Four patients showed a decrease in tumor mass, and five patients an unchanged tumor size. All had evidence of necrosis in post-TARE imaging, and all underwent LT. In the long term, no patient developed recurrence or died after a mean follow-up time of 15.8 months.

Ettorre et al. reported 3 patients being within the Milan criteria and 19 patients outside the Milan criteria for bridging and downstaging, respectively [25]. Initially, 41 patients had a downstaging intent and 9 dropped out before LT (22%). Eleven additional local therapies were necessary before LT. Downstaging to within the Milan criteria on imaging was achieved in 15 patients (78.9%) and histologically in 13 patients (59%). Eighteen patients (81.8%) had a complete or partial response after TARE on imaging. Progression-free survival after 5 years was approximately 90% and overall survival was approximately 70%. In the largest series reported, of the 93 patients who underwent LT, 25% of patients had more than one TARE [26]. Radiological assessment was only available for 88 patients, 31 patients (35.2%) were downstaged and 55 (62.5%) bridged while two progressed. Eight patients (9%) developed a recurrence after a median of 15.9 months. Median recurrence-free survival was 79 months; overall survival from TARE was not reached, but 57% were alive after 100 months.

### 3.4. Pooled Analyses

The studies reporting LT after TARE, radiologic or histologic response were included for the pooled analysis. The results are depicted in Figure 3. The pooled median survival after LT was 7.77 years (heterogeneity 25.5%) and the 5 year survival was 69.4% (95% CI 50.1–83.6).

### 3.5. Summary of TARE Effects on HCC

The effect of TARE on HCC was summarized by extracting data from each of 14 studies. The efficacy of TARE on tumors was 26.7%/49.2%/13.4%/10.7% for CR/PR/SD/PD in the downstaging group, 41.8%/34.1%/17.6%/6.6% in the bridging group, and 30.8%/35.9%/33.3%/0.0% in the mixed group (Figure 4). The downstaging rate by TARE was 45.9% and 38.8% in the downstaging and mixed groups, respectively, and the tumor regrowth rate after TARE was 42.5%, 14.7%, and 1.8% in the downstaging, bridging and mixed groups, respectively. PD rates and tumor recurrence rates tended to be higher in the downstaging group, which treated advanced cancer. 

For HCC with portal vein thrombosis (PVT), four of the fourteen studies reported on the therapeutic efficacy and prognosis of TARE; Pracht et al. reported that TARE for HCC patients with PVT resulted in a reduction of PVT in 83.3% of cases [14]. Serenari et al. reported that 30% of patients with PVT achieved CR and successful downstaging [18]. On the other hand, in the study of Gabr et al., only one of five UNOS 4b cases with PVT obtained downstaging to UNOS 3 and complete patency of the portal vein [26]. Mantry et al. also reported that both OS and PFS after TARE were lower in PVT cases than in non-PVT cases [20].

### 3.6. Comparison of TARE and other Treatments

Four comparative studies assessed outcomes in patients after TARE before LT or as a downstaging procedure [17,19,22,26] (Appendix A). One study presents a higher tumor response with the downstaging rate for liver transplant of TARE being superior to TACE (9/32 vs. 4/34) [19]. In the other three studies, the tumor response as assessed by either downstaging or histologic assessment was comparable. The posttransplant survival was comparable in both groups. Gramenzi and colleagues compared a matched group of patients who had either TARE or Sorafenib [15]. While survival was similar, only TARE treatment enables downstaging to within transplant criteria although this was only possible in 2/63 (3.2%). Ettorre et al. compared TARE before LT and no TARE before LT [25]. Patients who had TARE showed a lower survival up to 60 months after LT, but this was not statistically different for the observed period. Dhondt et al. described a median superior OS with censoring for orthotopic LT of 27.6 vs. 15.6 months comparing TARE and TACE in a randomized trial that was prematurely terminated for efficacy after enrolling 72 of 140 patients [19].

### 3.7. Adverse Events

Table 3 is showing the summary of adverse events (AE) after TARE. Six studies reported AEs [14,15,18,19,20,22]. One treatment-related death was observed at 87 days since the last treatment. Frequent AE were fatigue/asthenia, fever, gastrointestinal symptoms including transient liver dysfunction and pneumonia. The overall AE rate was between 21 and 60% and grade 3 or 4 AEs were detected between 15 and 30% of patients [15,18,20]. With a 90-day follow-up, the number of AEs seems to increase compared to shorter follow-up durations, but only two studies specified the follow-up duration for AEs [20].

## 4. Discussion

This study has shown that the experience with TARE is limited to a few expert centers. TARE is often used in conjunction with other therapies to control HCC confined to the liver. Whole liver procedures are seldom performed, and repeated interventions can be safely performed. The tumor control rate and response are high and long-term oncologic outcome after LT, although scarcely reported, is favorable as demonstrated in a pooled analysis. The safety profile is acceptable and often better compared to other local and systemic therapies. Procedure-related complications did not seem to lead to removal from the waiting list. Due to heterogeneous studies, no pooled analysis was conducted.

The best treatment option for HCC located exclusively in the liver is transplantation. Unfortunately, the scarcity of organs obliges us to make a selection of the best candidates for transplantation in order to achieve the best distribution of the available grafts. On the other hand, in recent years, there have been a series of advances in the field of LT. First, related to the eradication of the hepatitis C virus [27], grafts from donors with expanded criteria became available. Second, the use of pre-procurement normothermic regional perfusion was implemented to control donation after circulatory death [28]. In addition, there was the eruption of perfusion machines [29] and improvement in the results of the living donor [30], which has increased the number of available liver grafts worldwide.

This development has indirectly led to an expansion of the oncological indications for LT, which has been popularized by the concept of transplant oncology [31]. In the specific case of HCC, there has been a historical evolution of the criteria for LT. Bridging procedures are neoadjuvant therapeutic options that prevent disease progression during the waiting time and drop out of patients with HCC from the waiting list. Downstaging, if extending beyond the Milan criteria, shows similar results as LT for tumors already within the Milan criteria [32]. There are many options for locoregional therapies (either bridging or downstaging) including chemotherapy, transcatheter arterial chemoembolization (TACE), molecular therapy (targeted therapy, immunotherapy, or a combination of both), TARE with yttrium-90 microspheres or combined locoregional systemic therapy. At present, the studies about this broad array of interventions are very heterogeneous and there is not enough evidence in the literature to recommend one of these techniques over the others and often the preference for one over the other technique has developed historically. Therefore, an individualized selection of each of them will have to be made according to each patient and the logistics of each center.

The usefulness of TARE as a treatment has been endorsed by different authors and recommended by a multidisciplinary working group in terms of its safety, efficacy and feasibility for the surgical resection of patients with borderline-resectable HCC [33]. In contrast, in the field of LT, the role of TARE is under debate and there are still no clear and unified guidelines for its indication. Classically, TACE is the treatment of choice for downstaging and bridging for HCC but there is a growing trend in favor of TARE, especially in intermediate stage or in unresectable HCC due to tumor size or the number of nodules [34,35]. Kim et al. have described that TARE has a higher disease control rate, significantly better overall and intrahepatic PFS, and better survival outcomes in patients without lymph node or distant metastases, with BCLC stage B or C and a tumor size ≥5 cm [36]. Recently, results from the chemoembolization-controlled phase II TRACE trial concluded that with a similar safety profile, chemoembolization with yttrium-90 conferred superior tumor control and survival compared to chemoembolization in selected participants with early or intermediate HCC. The authors downstaged 10 patients in the TARE arm and 4 in the DEB-TACE arm to transplant with a higher median overall survival for TARE than TACE in liver transplant patients. In addition, a subgroup analysis of participants with BCLC stage B HCC treated with TARE showed a longer time to overall tumor progression (12.8 vs. 9.6 months).

The present review and current evidence from the literature analyzing the role of TARE vs. TACE fuel the debate on whether patients with HCC with a large lesion between 5 and 8 cm in size or with several nodules between 3 and 5 cm (especially if they are in the same hepatic lobe) and who are not surgical candidates or are outside the established criteria for transplantation may benefit from a TARE (Figure 5). The main argument supporting this hypothesis would be that it allows greater control of tumor progression while on the waiting list with a tumor response that allows sufficient downstaging to include the patient for LT within established criteria. Appropriate patient selection, the biological behavior of tumours and the individual situation of each transplant center in terms of its available therapeutic armamentarium and its management of the waiting list are key to giving TARE its appropriate place in LT.

Some limitations require attention. In all studies, there is a baseline selection of patients not suitable for locoregional therapy. In addition, the follow-up assessment was prone to further selection due to high dropouts for various reasons, the most common being death or tumor progression. In addition, all comparison is most likely underpowered, and a type II error cannot be excluded. Pooled analyses were based on heterogeneous protocols and populations and should be interpreted cautiously. Ultimately, we can only report findings of a selected subgroup of patients suitable for TARE and with available outcome data.

## 5. Conclusions

The use of TARE is an alternative in patients with advanced HCC not initially suitable for resection or LT. This locoregional therapy can be used especially in those patients with tumors outside the established criteria for LT but with possibilities of salvage due to the size or number of lesions. It also has the potential to be used safely in patients with portal vein thrombosis. If it meets established indication criteria and can save patients for LT, it could be a safe treatment option. However, the current evidence is very limited and future studies need to better define which patients could benefit from this therapy, especially in comparison to other established procedures such as TACE.

## Figures and Tables

**Figure 1 cancers-15-02122-f001:**
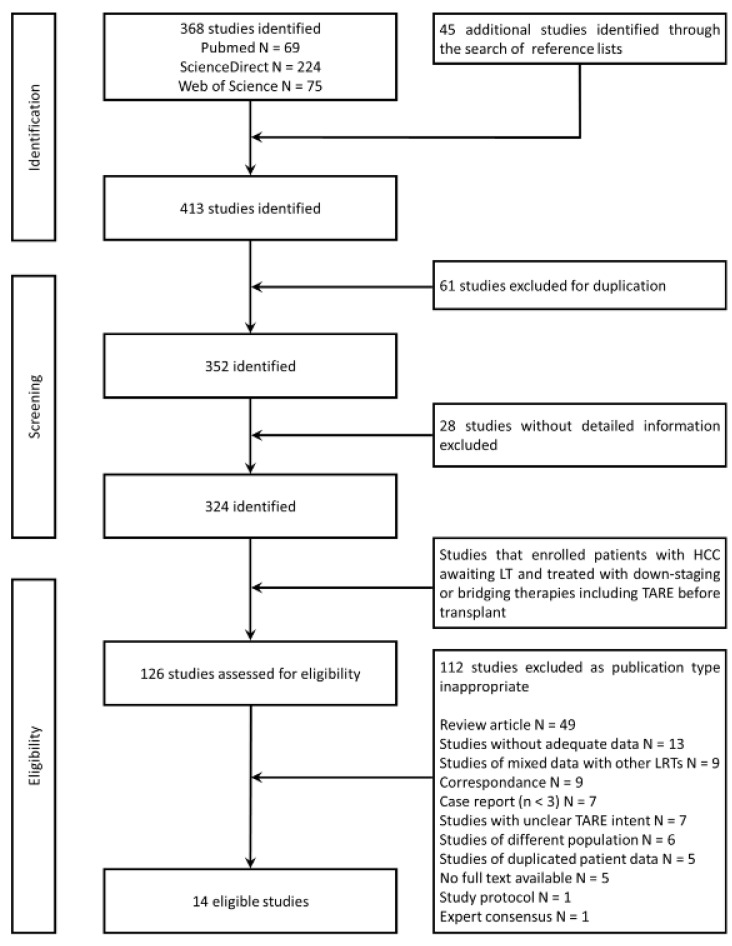
Flow chart of study selection following PRISMA (Preferred Reporting Items for Systematic reviews and Meta-Analyses). HCC: hepatocellular carcinoma, LT: liver transplant, TARE: transarterial radioembolization, LRT: locoregional therapy.

**Figure 2 cancers-15-02122-f002:**
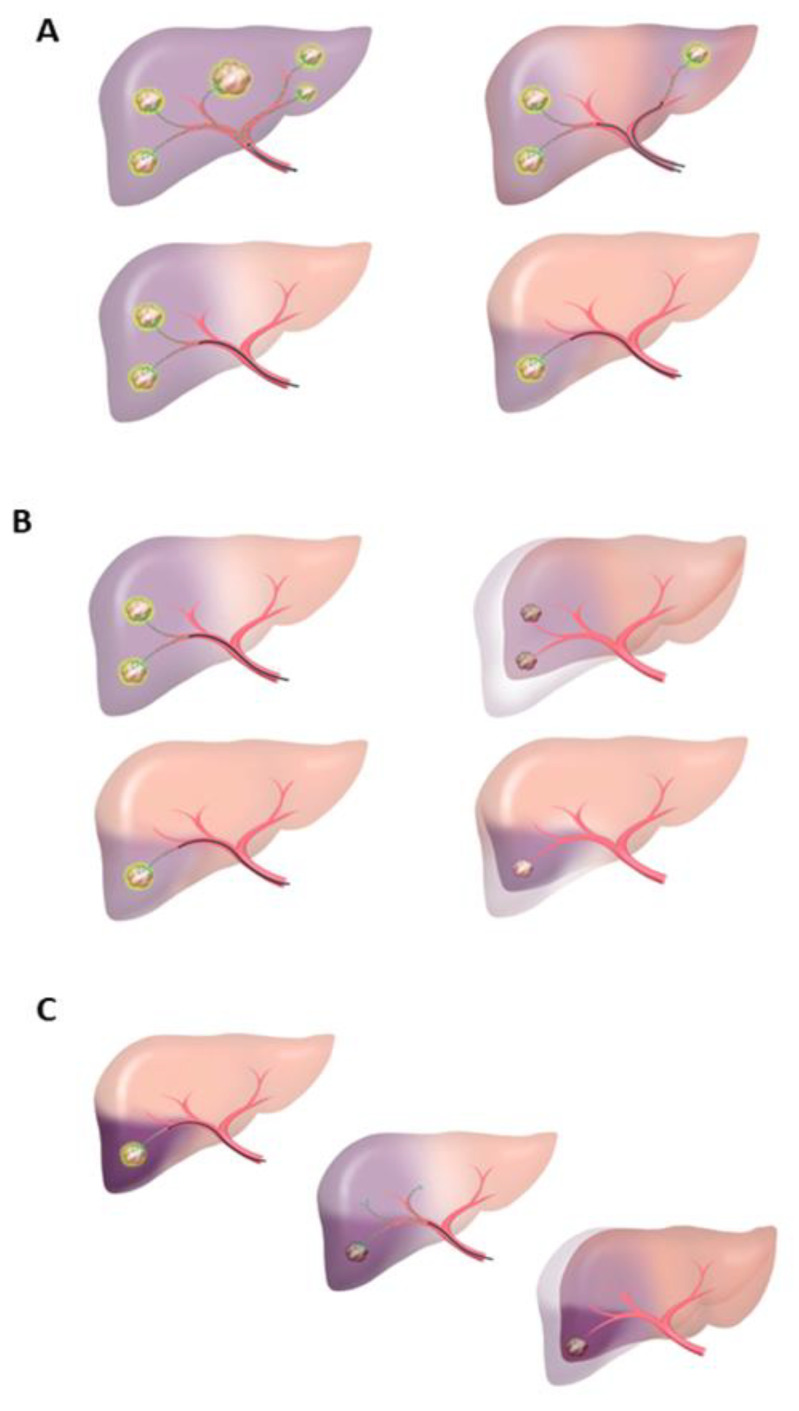
Various TARE procedures. (**A**) Types of TAREs: total (whole liver), bilobar (2 catheters), lobar or segmental and subsegmental; (**B**) Segmental TARE: before and after (healthy lobe hypertrophy and diseased segment atrophy); (**C**) TARE in 2 phases: boost in LOE and lower dose in the corresponding lobe. Finally, atrophy of the diseased lobe and hypertrophy of the healthy lobe.

**Figure 3 cancers-15-02122-f003:**
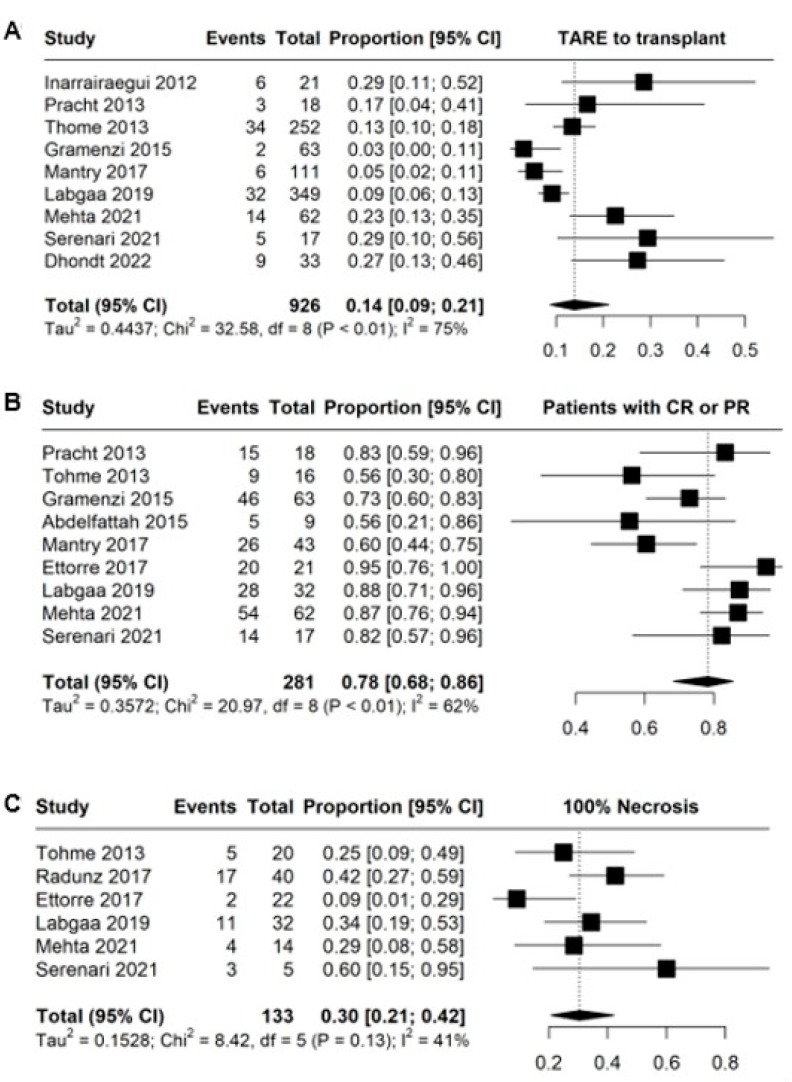
Results of pooled single-arm analyses. (**A**) Proportion of patients proceeding to transplant; (**B**) proportion of patients with either complete or partial response of the tumor; (**C**) proportion of patients with 100% tumor necrosis on histological assessment.

**Figure 4 cancers-15-02122-f004:**
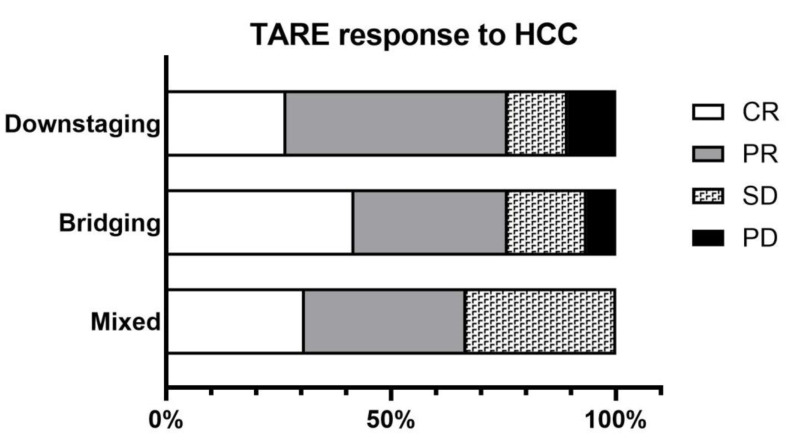
TARE response to HCC in 14 studies. Data on the effect of tumor treatment on TARE were extracted from each study and summarized: HCC: hepatocellular carcinoma, TARE: transarterial radioembolization, CR: complete response, PR: partial response, SD: stable disease, PD: progressive disease.

**Figure 5 cancers-15-02122-f005:**
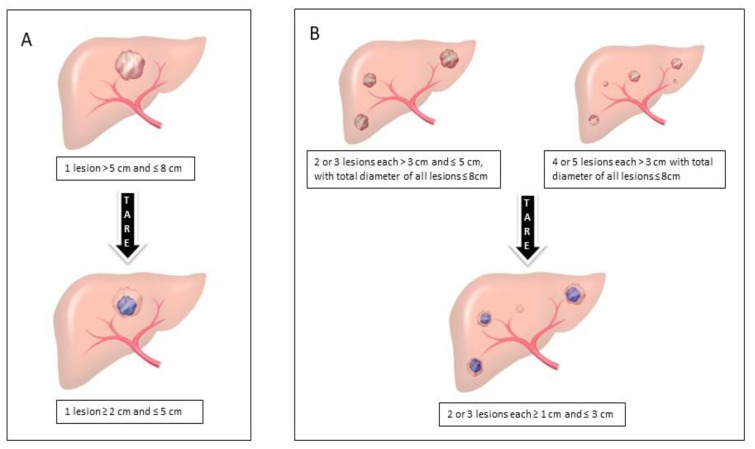
General aspects of the criteria for downstaging in patients with hepatocellular carcinoma who are candidates for liver transplantation. (**A**) Single lesion; (**B**)Two or more lesions.

**Table 1 cancers-15-02122-t001:** Baseline characteristics of patients in all studies at TARE.

Study	Period	Sex (Male/Female), *n*	Age, Years	Child-Pugh (A,B,C), *n*	MELD, Score	Etiology (HBV/HCV/ ALC/NASH/ Others), *n*	BCLC (0/A/B/C/D), *n*	Multifocal, *n*	Number of Nodules, *n*	Size of Largest Nodule, mm	AFP, ng/mL	Time from TARE to LT, Months
Downstaging												
Iñarrairaegui et al. [13]	2003–2010	17/4	72	-	-	-	0/12/9/0/0	5	16/4/1 (1/2/3)	80	12	22.5 (10–35)
Pracht et al. [14]	2007–2010	12/6	63 (44–77)	13/5/0	-	0/4/10/1/3	-	7	-	-	36.5 (3–91,000)	-
Gramenzi et al. [15]	2005–2012	28/4	18/4 (<70/≥70 years)	29/3/0	11/21 (<9/≥9)	-	0/0/15/17/0	27	-	-	13/12/7 (≤20/21–200/>200 ng/mL), *n*	-
Labgaa et al. [16]	2012–2016	15/7	61 (56–64)	0/8/13	31 (25–33)	2/15/2/0/3	0/19/3/0/0	14	-	-	-	-
Mehta et al. [17]	2016–2019	45/17	63 (60–66)	50/12/0	8.5 (7–10)	-	-	-	30/25/7 (1/2–3/4–5)	-	17.9 (5.7–238.4)	15.9 (11.2–19.2)
Serenari et al. [18]	2013–2016	15/2	53 (50–56)	15/2/0	-	1/12/0/3/1	-	-	1 (1–2)	59 (43–70)	18.6 (7.3–103.4)	24.9 (6.2–32.6)
Dhondt et al. [19]	2011–2018	28/4	68 (64–74)	30/2/0	-	0/5/21/1/5	0/5/27/0/0	25	17 (>3 nodules)	42 (32–56)	28/3/1 (<400/≥400/data missing ng/mL)	-
Bridging												
Mantry et al. [20]	2004–2013	85/26	65.8 ± 9.6 *	82/26/3	16/25/18/20/32 (6/7/8/9/≥10)	9/65/27/12/8	0/38/51/22/0	-	-	-	-	-
Radunz et al. [21]	2007–2015	32/8	59 ± 6 *	-	12 (6–40)	8/9/12/0/11	3/15/4	25	-	35 (5–110)	22.5 (1–13,926)	4.2 (0.4–21.6)
Zori et al. [22]	2012–2017	21/7	23/5 (<65 ≥ 65)	-	13.5 **	2/19/1/0/6	-	17	-	-	121 **	10.1 **
Mixed												
Tohme et al. [23]	2001–2011	16/4	60.2 ± 6.8 *	-	13 ± 8 *	3/8/4/0/5	-	-	9/7/4 (1/1–3/>3)	-	17 **	3.5 **
Abdelfattah et al. [24]	-	4/5	53.8 ± 9.5 *	-	-	1/5/0/0/3	0/12/9/0/0	-	-	50 (10–87)	13 (5–499)	15.8 ± 17.7 *
Ettorre et al. [25]	2002–2015	22/0	55 (41–67)	-	-	2/17/2/1/0	0/3/15/4/0	4	2 (0–13)	18.5 (0–60)	-	14.5 (2–60)
Gabr et al. [26]	2003–2013	24/69	60 (57–64)	47/42/4	-	11/47/14/6/15	2/62/16/9/4	33	-	-	18.4 (5.1–250.3)	6.5 (3.7–9.9)

Values are expressed as median (range), or number (*n*) as indicated. * Values with means ± SD (standard deviation). ** Values with means. TARE: transarterial radioembolization, LT: liver transplant, MELD: Model for End-stage Liver Disease, HBV: hepatitis B virus, HCV: hepatitis C virus, Alch: alcoholic, NASH: nonalcoholic steatohepatitis, BCLC: Barcelona Clinic Liver Cancer staging system, AFP: alpha-fetoprotein.

**Table 2 cancers-15-02122-t002:** Oncological outcomes after LT following TARE.

Study	TARE → LT, *n*	Reccurence, *n* (%)	Median PFS, Months	Median OS, Months	1 Year OS, % and 95% CI	2 Year OS, % and 95% CI	3 Year OS, % and 95% CI	5 Year OS, %
Downstaging								
Iñarrairaegui et al. [13]	21 → 2	0 (0.0)	-	27 (5.0–48.9) *	100	100	100	-
Pracht et al. [14]	18 → 1	-	50.6, [11.0 (8.0–16.5) *]	16.0	70.3 ± 21.1 *^‡^	-	-	-
Gramenzi et al. [15]	32 → 2	18 (41.9) *	-	11.2 (6.7–15.7) *	44.7 *	19.0 *	9.5 *	-
Labgaa et al. [16]	22 → 22	0 (0.0)	-	-	100	95	91	91
Mehta et al. [17]	62 → 14	5 (7.9)	16.8 (9.7–22.3) ^†^	-	100	95.0	83.1	-
Serenari et al. [18]	17 → 5	3 (60.0)	34.6 (10.9–58.2)	-	80	-	80	60
Dhondt et al. [19]	32 → 9	-	17.1 (6.5–27.8) *	30.2 (20.4–39.9) *	81.3 *	56.3 *	21.9 *	6.3 *
Bridging								
Mantry et al. [20]	111 → 6	-	9.8 (6.8–14.8) *	69.0, [13.1 (10.3–18.4) *]	46.8 *	26.1 *	10.8 *	-
Radunz et al. [21]	40 → 40	9 (22.5)	13 (4–56) ^†^	46	77.5	-	-	50
Zori et al. [22]	28 → 28	-	16.8 †	-	96.4 ± 3.5 ^‡^	96.4 ± 3.5 ^‡^	92.9 ± 4.9 ^‡^	-
Mixed								
Tohme et al. [23]	20 → 20	4 (20.0)	36.8 (9.4–62.1) ^†^	75.1 (36.9–106.0)	95	84	-	79
Abdelfattah et al. [24]	9 → 9	0 (0.0)	-	-	-	-	-	-
Ettorre et al. [25]	22 → 22	-	29.6 (mean)	30.2 (mean)	-	-	-	-
Gabr et al. [26]	93 → 93	8 (9.0)	15.9 (7.8–46.8) ^†^	57% (OS at 100 months)	-	-	-	67 *

TARE: transarterial radioembolization, LT: liver transplant, PFS: progression free survival, OS: overall survival, AFP: alpha-fetoprotein. Values are expressed as median (range), or number (*n*) as indicated. * Values with all TARE patients. ^†^ Median time from LT to recurrence. ^‡^ Values with means ± SD (standard deviation). TARE: transarterial radioembolization, LT: liver transplant, PFS: progression free survival; OS: overall survival; CI: confidence interval.

**Table 3 cancers-15-02122-t003:** Adverse events after TARE.

	Overall AE Rates	Minor AEs	Severe AEs	Treatment Related Death
Pracht et al. [14]	27.80%	Ascites (27.8%)	-	-
Gramenzi et al. [15]	59%	Fatigue (9%), fever (6%), nausea/vomiting (2%), abdominal pain (8%), Child–Pugh score deterioration (21 patients).	Radiation pneumonia (8%), RILD (8%), cholecystitis (5%).	-
Serenari et al. [18]	23.50%	Grade 1 abdominal pain in 3 patients, and grade 1 fever and fatigue in 1 patient, mild ascites in 2 patients.	-	-
Dhondt et al. [19]	39%	-	Renal and urinary disorders (15%), hepatobiliary disorders (42%), RILD (3%)	3%
Mantry et al. [20]	43.40%	Abdominal pain (14.2%), ascites (18.9%), nausea (5.7%), edema (6.6%), fatigue (2.8%), vomiting (4.7%).	Ulcer (4.7%), jaundice (3.8%), GI bleeding (7.5%)	-
Zori et al. [22]	1.70%		Hyperbilirubinemia (1.7%)	-

AE: adverse event, RILD: radiation-induced liver disease.

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
