# Peer review of "Selecting the Appropriate Downstaging and Bridging Therapies for Hepatocellular Carcinoma: What Is the Role of Transarterial Radioembolization? A Pooled Analysis"

_cancers, 2023, doi:10.3390/cancers15072122_

Round 1

Reviewer 1 Report

Dr Lopez Lopez and co-authors present a systematic review aiming to summarise the current role of TARE as a downstaging or bridging therapeutic strategy in patients with HCC awaiting LT.

They state that TARE seems a safe therapeutic option and presents potential advantages in its capacity to reduce the size of the lesion and the toxic effect.

Overall, the idea is good and the review is quite comprehensive. However, there are methodological and clinical issues I’d like to highlight.

Major issue:

Despite the review being quite comprehensive I have the feeling that it somehow lacks a synthesis of key findings, especially in the first 3 paragraphs of the results section. The authors describe in detail important findings from each study, but it’s hard to find key findings that support the conclusions. It would be helpful for the reader to have a more structured results section highlighting the successful rate of TARE, the complication rate of TARE, outcomes of TARE in case of PVTT, etc.. . Additional analysis, figures or tables could be helpful. More comments follow below.

Methods and results:

- Is there an available study protocol (online published)?

- The RoB2 tool should be used only for RCT. How did you assess the quality of observational studies? This should be specified and discussed.

- The quality assessment paragraph is reported twice (duplicated). Please delete it.

- Please provide a definition of whole live TARE.

- Similarly, information on how TARE was generally performed would be helpful for the reader.

- Figures summarising the % of effective/not-effective TARE, downstaged/not-downstaged HCC, and recurrence after TARE (in absence of LT) would be insightful for the reader. A figure on articles assessing the effect of TARE on patients with HCC and PVTT could be interesting for the reader.

- Even if there are just a few studies (with significant clinical heterogeneity) I’d encourage the authors to perform a quantitative review (meta-analysis). Alternatively, a pool data analysis could be performed in order somehow summarise data, thus providing more solid figures supporting the conclusions of the paper.

- It could be useful to expand the section “comparative studies”. A table could be helpful. What was the TARE compared with? What were the outcomes of these comparisons? What were the baseline characteristics of the study and control group?

- Similarly, I’d encourage the author to expand the “adverse events” paragraph. Again, a table summarising key findings could be helpful as well.

Discussion and conclusions:

Conclusions should be softened because of: 1) the quality of evidence from included studies and 2) they are not supported by an appropriate analysis (i.e. “tumours outside the established criteria for LT but with possibilities of salvage due to the 356 size or number of lesions”)

Other minor issues:

- There are several language typos. Additional proofreading by a native/professional proofreader is recommended (i.e. “Only RCT included 39 presents…” – “TARE for downstaging or bridging in LT seems that it is a…”- etc..)

- When citing an article, it would be recommendable to prefer the form “author et al.” instead of “author” (i.e. Mehta and Dhondt)

Author Response

Dr Lopez Lopez and co-authors present a systematic review aiming to summarise the current role of TARE as a downstaging or bridging therapeutic strategy in patients with HCC awaiting LT. They state that TARE seems a safe therapeutic option and presents potential advantages in its capacity to reduce the size of the lesion and the toxic effect. Overall, the idea is good and the review is quite comprehensive. However, there are methodological and clinical issues I’d like to highlight.

Major issue:

Despite the review being quite comprehensive I have the feeling that it somehow lacks a synthesis of key findings, especially in the first 3 paragraphs of the results section. The authors describe in detail important findings from each study, but it’s hard to find key findings that support the conclusions. It would be helpful for the reader to have a more structured results section highlighting the successful rate of TARE, the complication rate of TARE, outcomes of TARE in case of PVTT, etc.. . Additional analysis, figures or tables could be helpful. More comments follow below.

Reply: We would like to thank the Reviewer for giving us constructive comments which significantly improved our manuscript. According to the Reviewer’s suggestion, we have performed additional validation to demonstrate clinical significances and more detailed results as described below.

Methods and results:

- Is there an available study protocol (online published)?

Reply: Yes, we have. The study protocol was registered in PROSPERO (registration number: CRD42023383661). We have added following statement regarding PROSPERO registration to the Methods section. The study protocol was registered in PROSPERO (registration number: CRD42023383661).

- The RoB2 tool should be used only for RCT. How did you assess the quality of observational studies? This should be specified and discussed.

Reply:

- The quality assessment paragraph is reported twice (duplicated). Please delete it.

Reply: Thank you for pointing out this important error. We have removed the following duplicate statement.

- Please provide a definition of whole live TARE.

Reply:  This is shown in Figure 2. Maybe the figure needs more detailed guidance e.g. under “A” there are four figures and one shows whole liver TARE but we cannot refer to this specific figure. We discussed this simultaneously in the following response to comments on TARE's methodology.

- Similarly, information on how TARE was generally performed would be helpful for the reader.

Reply: The following new paragraphs have been added to the materials and methods section on TARE methodology. The types of TAREs were also discussed in this paragraph.

TARE procedure

Treatment commonly reported as TARE was defined as including the following procedures. In the TARE technique, microspheres impregnated with a radioisotope yttrium-90 are percutaneously and selectively delivered through the hepatic artery to targeted tumor. Yttrium-90 is a beta-ray emitter with a short half-life and limited tissue penetration. Normally, microspheres are available in glass or resin, which differ in size, activity of individual beads, and number of microspheres injected. Tumor mapping with angiography using cone-beam CT and treatment simulation with 150 MBq technetium-99m macro-aggregated albumin are usually performed prior to treatment. After treatment, SPECT/CT or PET/CT will also be performed to assess microsphere distribution and mean dose to the target, liver, and lungs.

TARE methods include whole liver TARE, in which microshpere is spread from the proper hepatic artery to the entire liver, bilobar TARE, in which catheters are inserted into both lobes simultaneously, and lobar or segmental and subsegmental TARE, which targets one lobe or less. In addition, a two-phase TARE is performed to promote gradual atrophy of the responsible liver lobe and enlargement of the healthy liver lobe by applying the partial atrophy of the liver caused by segmental TARE (Figure 2).

- Figures summarising the % of effective/not-effective TARE, downstaged/not-downstaged HCC, and recurrence after TARE (in absence of LT) would be insightful for the reader. A figure on articles assessing the effect of TARE on patients with HCC and PVTT could be interesting for the reader.

Reply: We have added following paragraph and figure regarding to the effect of TARE on HCC summarizing 14 reports.

Summary of TARE effects on HCC

The effect of TARE on HCC was summarized by extracting data from each of 14 studies. The efficacy of TARE on tumors was 26.7%/49.2%/13.4%/10.7% for CR/PR/SD/PD in the downstaging group, 41.8%/34.1%/17.6%/6.6% in the bridging group, and 30.8%/35.9%/33.3%/0.0% in the mixed group (Figure 4). The downstaging rate by TARE was 45.9% and 38.8% in the downstaging and mixed groups, respectively, and the tumor regrowth rate after TARE was 42.5%, 14.7%, and 1.8% in the downstaging, bridging, and mixed groups, respectively. PD rates and tumor recurrence rates tended to be higher in the downstaging group, which treated advanced cancer.

For HCC with portal vein thrombosis (PVT), four of the 14 studies reported on the therapeutic efficacy and prognosis of TARE; Pracht et al. reported that TARE for HCC patients with PVT resulted in reduction of PVT in 83.3% of cases (14). Serenari et al. reported that 30% of patients with PVT achieved CR and successful downstaging (18). On the other hand, in the study of Gabr et al., only one of five UNOS 4b cases with PVT obtained downstaging to UNOS 3 and complete patency of the portal vein (26). Mantry et al. also reported that both OS and PFS after TARE were lower in PVT cases than in non-PVT cases (20).

- Even if there are just a few studies (with significant clinical heterogeneity) I’d encourage the authors to perform a quantitative review (meta-analysis). Alternatively, a pool data analysis could be performed in order somehow summarise data, thus providing more solid figures supporting the conclusions of the paper.

Reply: We agree with the reviewer that the inclusion of a panel of forest plots figures could contribute to supports the conclusions of the paper. In this regard, we have added in the results section a single-arm pooled analysis of the proportion of patients proceeding to transplantation; proportion of patients with complete or partial tumor response; and proportion of patients with 100% tumor necrosis on histologic evaluation.

- It could be useful to expand the section “comparative studies”. A table could be helpful. What was the TARE compared with? What were the outcomes of these comparisons? What were the baseline characteristics of the study and control group?

Reply: To make it clear that TARE is being compared to TACE, sorafenib or other therapies; a table has been added as supplementary table 2 that corresponds to the content of the comparative studies.

- Similarly, I’d encourage the author to expand the “adverse events” paragraph. Again, a table summarising key findings could be helpful as well.

Reply: A table summarizing the types and incidence of adverse events described in each study has been added as Table 3 to correspond with the paragraph on adverse events.

Discussion and conclusions:

Conclusions should be softened because of: 1) the quality of evidence from included studies and 2) they are not supported by an appropriate analysis (i.e. “tumours outside the established criteria for LT but with possibilities of salvage due to the 356 size or number of lesions”)

Reply: Thank you for your important suggestion. We have changed the wording of the conclusion section to a softer wording that avoids assertions, as follows.

It also has the potential to be used safely in patients with portal vein thrombosis. If it meets established indication criteria and can save patients for LT, it could be a safe treatment option. However, the current evidence is very limited and future studies need to better define which patients could benefit from this therapy, especially in comparison to other established procedures such as TACE.

Other minor issues:

- There are several language typos. Additional proofreading by a native/professional proofreader is recommended (i.e. “Only RCT included 39 presents…” – “TARE for downstaging or bridging in LT seems that it is a…”- etc..)

Reply: We agree that the language should be edited in light of the fact that all authors are non-native English speakers and we have put much effort to correct the English in the manuscript

- When citing an article, it would be recommendable to prefer the form “author et al.” instead of “author” (i.e. Mehta and Dhondt)

Reply: In accordance with the recommendation, we have modified the format of the authorship of the paper.

Reviewer 2 Report

I thank you for the opportunity to re view the article with the title “Selecting the appropriate downstaging and bridging therapies 2 for HCC: what is the role of transarterial radioembolization?” by Victor Lopez-Lopez and colleagues. The authors performed a systematic review of the literature on the role of transarterial radioembolization (TARE) in liver resection and liver transplantation. I have the following comments.

1.       Sections 2.2 and 2.3 are redundant.

2.       Table 1 needs to be further refined making the data easier to read and follow.

3.       Table 2 is good and relevant.

4.       Table 3 reflects little effort from the authors to consolidate the studies and present the indications in a more clinically useful way.

5.       On the lie 296, I believe the authors meant “eruption” rather than irruption.

6.       The abstract conclusion “TARE for downstaging or bridging in LT seems that it is a safe therapeutic option and 42 presents potential advantages in its capacity to reduce the size of the lesion and the toxic effect 43 resulting in necrosis”, perhaps could be corrected making the statement more definitive.

Overall, the manuscript has useful information but at the end of the review, the data were not presented in a way that directly and convincingly support the conclusion. I highly recommend that the outcomes of TARE should be presented using Forest’s plots illustrating visually the results associated with TARE compared to historical or TACE in liver transplant and liver resection.

Author Response

Please correct me, Thank you for the opportunity to re view the article with the title “Selecting the appropriate downstaging and bridging therapies 2 for HCC: what is the role of transarterial radioembolization?” by Victor Lopez-Lopez and colleagues. The authors performed a systematic review of the literature on the role of transarterial radioembolization (TARE) in liver resection and liver transplantation. I have the following comments.

  1. Sections 2.2 and 2.3 are redundant.

Thank you for pointing out this important error. We have removed the following duplicate statement.

  1. Table 1 needs to be further refined making the data easier to read and follow.

We agree with the reviewer's comment and in order to improve the understanding of the table we have removed the TARE LT and Downstaging/Bridging rows.

  1. Table 2 is good and relevant.

Thank you for your comment.

  1. Table 3 reflects little effort from the authors to consolidate the studies and present the indications in a more clinically useful way.

We understand the reviewer's comment and share his thoughts. We have decided to remove the table from the main text and attach it as a supplementary table because, as the reviewer points out, it exclusively reflects the indications of each center. The idea is that due to the heterogeneity of the use of TARE in the field of liver transplantation, we wanted to summarize the criteria used by each group to better understand the results of the review.

  1. On the lie 296, I believe the authors meant “eruption” rather than irruption.

Thank you for pointing out this grammar confusion. We have changed it in the manuscript.

  1. The abstract conclusion “TARE for downstaging or bridging in LT seems that it is a safe therapeutic option and 42 presents potential advantages in its capacity to reduce the size of the lesion and the toxic effect 43 resulting in necrosis”, perhaps could be corrected making the statement more definitive.

We have followed the reviewer's recommendations by modifying the message of the conclusion to be more forceful within the current evidence.

Overall, the manuscript has useful information but at the end of the review, the data were not presented in a way that directly and convincingly support the conclusion. I highly recommend that the outcomes of TARE should be presented using Forest’s plots illustrating visually the results associated with TARE compared to historical or TACE in liver transplant and liver resection.

We appreciate your comments because they have helped to reorganize both the tables and the text and to improve their comprehensibility. In order to improve the comprehensibility of the message of the review we have added a comparison of TARE and other treatments, a summary of the effects of TARE in HCC, and a pooled analysis reporting LT after TARE, radiologic or histologic response, and oncologic outcomes

Reviewer 3 Report

the review is very important to the field of HCC management with well followed scientific approach and proper writing style, it can be published as it is 

Author Response

We highly appreciate the reviewer’s comment and thank you for the time taken to critically assess our study.

Round 2

Reviewer 2 Report

Thank you for sending me this final version. I find the revision acceptable and I support accepting the manuscript.